# Highly Selective Photocatalytic Reduction of o-Dinitrobenzene to o-Phenylenediamine over Non-Metal-Doped TiO$_2$ under Simulated Solar Light Irradiation

**Hamza M. El-Hosainy** [3,4]**, Said M. El-Sheikh** [1]**, Adel A. Ismail** [1,2,]*****, Amer Hakki** [3]**, Ralf Dillert** [3] **, Hamada M. Killa** [5]**, Ibrahim A. Ibrahim** [1] **and Detelf W. Bahnemann** [3,6,]*****

1   Department of Nanomaterials and Nanotechnology, Central Metallurgical R & D Institute,
    Cairo 11421, Egypt; selsheikh2001@gmail.com (S.M.E.-S.); ibrahimahmedcmrdi01@gmail.com (I.A.I.)
2   Nanotechnologyand and Advanced Materials Program, Energy & Building Research Center,
    Kuwait Institute for Scientific Research (KISR), P.O. Box 24885, Safat 13109, Kuwait
3   Institut für Technische Chemie, Leibniz Universität Hannover, Callinstr. 3, D-30167 Hannover, Germany;
    hamzaelhosainy@gmail.com (H.M.E.-H.); a.hakki@abdn.ac.uk (A.H.); dillert@iftc.uni-hannover.de (R.D.)
4   Institute of Nanoscience & Nanotechnology, Kafrelsheikh University, Kafrelsheikh 33516, Egypt
5   Faculty of Science, Zagazig University, Zagazig 44519, Egypt; hamadakilla48@gmail.com
6   Laboratory "Photoactive Nanocomposite Materials", Saint-Petersburg State University, Ulyanovskaya str. 1,
    Peterhof, Saint-Petersburg 198504, Russia
*   Correspondence: aaismail@kisr.edu.kw (A.A.I.); bahnemann@iftc.uni-hannover.de (D.W.B.)

**Abstract:** Photocatalytic reduction and hydrogenation reaction of o-dinitrobenzene in the presence of oxalic acid over anatase-brookite biphasic TiO$_2$ and non-metal-doped anatase-brookite biphasic TiO$_2$ photocatalysts under solar simulated light was investigated. Compared with commercial P25 TiO$_2$, the prepared un-doped and doped anatase-brookite biphasic TiO$_2$ exhibited a high selectivity towards the formation of o-nitroaniline (85.5%) and o-phenylenediamine ~97%, respectively. The doped anatase-brookite biphasic TiO$_2$ has promoted photocatalytic reduction of the two-nitro groups of o-dinitrobenzene to the corresponding o-phenylenediamine with very high yield ~97%. Electron paramagnetic resonance analysis, Transient Absorption Spectroscopy (TAS) and Photoluminescence analysis (PL) were performed to determine the distribution of defects and the fluorescence lifetime of the charge carriers for un-doped and doped photocatalysts. The superiority of the doped TiO$_2$ photocatalysts is accredited to the creation of new dopants (C, N, and S) as hole traps, the formation of long-lived Ti$^{3+}$ defects which leads to an increase in the fluorescence lifetime of the formed charge carriers. The schematic diagram of the photocatalytic reduction of o-dinitrobenzene using the doped TiO$_2$ under solar light was also illustrated in detail.

**Keywords:** photocatalysis; non-metal- doped TiO$_2$; nitroaromatic compounds; reduction; selectivity

## 1. Introduction

The great challenge for the modern chemical industry is to drive chemical reactions employing a sustainable, green, and eco-friendly process using renewable energy sources. Therefore, the development of new strategies to obtain fine chemicals in a fast, clean, and efficient approach is of great significance and still requires considerable efforts. Heterogeneous photocatalysis is one of the promising and eco-friendly approaches that satisfies these requirements. This method is one of the important processes that encourages the use of sunlight as the source for chemical conversion processes [1–4]. Moreover, the photo-induced organic transformations by solar-driven

photocatalysis can produce specific products with high selectivity and lower cost compared to conventional methods [5–7]. The reduction of nitroaromatic compounds to the corresponding amino compounds is one of these photo-induced organic transformations with $TiO_2$ is [3–6] which has attracted significant attention because of the importance of these amino compounds as intermediates of numerous valuable compounds such as dyes and medicines [8]. Most of the reported studies in this field have used alcohols as reaction media, hole scavenger, and hydrogen source for photocatalytic hydrogenation of nitroaromatics using $TiO_2$ photocatalyst under inert gas atmosphere [9,10]. However, the oxidation products of alcohols may react with the reduction product of the nitro compounds which affect the selectivity and the yield of the desired amino compound. Thus, of other hole scavengers, oxalic acid is preferred since it is easily dissolved in water, which can be used as reaction media, and $CO_2$ is the only byproduct of its oxidation [4,11–13]. $TiO_2$ (P-25) was employed as a photocatalyst for the photoreduction of dinitro compound o-dinitrobenzene to corresponding mono and diamino compounds (o-nitroaniline and o-phenylenediamine) under solar light irradiation [14]. The results showed formation of low yield and selectivity of 55% o-nitroaniline and 30% o-phenylenediamine.

On the other hand, the reduction of nitroaromatic compounds by non-metal-doped $TiO_2$ under solar light is rarely investigated. One example is the use of N-doped $TiO_2$ together with KI for the reduction of o-nitrophenol in the existence of methanol under solar light [3]. The results revealed that N-$TiO_2$ has a low efficiency for reduction of nitroaromatic compounds under solar light irradiation. The phase type of $TiO_2$ also has a great effect on the photocatalytic selectivity. Rutile $TiO_2$ displays higher activity and selectivity than anatase $TiO_2$ and P25 $TiO_2$ for selective hydrogenation of nitroaromatic compounds [15]. Furthermore, compared with rutile $TiO_2$, P25 $TiO_2$ has the ability to complete photocatalytic reduction of m-dinitrobenzene to the corresponding m-phenylenediamine in the deaerated aqueous iso-propanol under 4 h of UV light irradiation [16]. Herein, we report for the first time the use of anatase-brookite biphasic $TiO_2$ and non-metal (C, N and S)-doped anatase-brookite biphasic $TiO_2$ for the selective hydrogenation of o-dinitrobenzene to the corresponding o-nitroaniline and o-phenylenediamine under solar simulator light, respectively. To the best of our knowledge, there are no reports showing the high selectivity and formation of o-phenylenediamine by using non-metal-doped anatase-brookite biphasic $TiO_2$ under solar light irradiation. The expected schematic diagram and mechanism for the photocatalytic reduction of o-dinitrobenzene was also interpreted.

## 2. Results and Discussion

Similar to our previous work [17], XRD analysis proved the formation of anatase and brookite biphase $TiO_2$ with compositions ~75% and 25%, respectively. The surface area and pore size for un-doped and doped samples amounted to 226.2 and 85.1 $m^2\ g^{-1}$ and 2.2 nm and 3.6 nm, respectively. Therefore, these results show the prepared $TiO_2$ samples have a mesoporous structure. The particle size for un-doped and doped samples were 5–10 nm and 10–15 nm, respectively. The XPS analysis has proved the existence of C, N, S in the doped sample. UV-Vis. spectroscopy displayed a red absorption shift for the doped sample, reflecting that the band gap value of doped $TiO_2$ sample decreased from 3.1 to 2.9 eV. Herein, the feasibility of using these materials for photocatalytic reduction of o-dinitrobenzene in aqueous oxygen-free solutions under solar light irradiation was also conducted.

*2.1. Reduction of O-Dinitrobenzene to O-Nitroaniline and O-Phenylenediamine*

Firstly, initial experiments for reduction of o-dinitrobenzene were carried out with an aqueous solution containing $TiO_2$ samples in the existence of oxalic acid as a hole scavenger.

Figure 1 represents the time-dependent change in the concentration of o-dinitrobenzene and its photocatalytic products in its aqueous solution containing either un-doped (a) or doped $TiO_2$ (b) in presence of oxalic acid as hole scavenger during the irradiation with solar simulated light. It is clearly observed that the concentration of o-dinitrobenzene dramatically decreases with increasing of the photoirradiation time for both T and DT samples. 9 h were needed to achieve the complete conversion of o-dinitrobenzene when employing T as the photocatalyst, whereas only 7 h were enough in the case

of DT sample. Interestingly, the concentration of corresponding monoamino compound (o-nitroaniline) increases gradually with prolonged photoirradiation time when employing the un-doped photocatalyst (T). However, the photo-catalytically produced (o-nitroaniline) undergoes further reduction and hydrogenation to produce the corresponding diamino product (o-phenylenediamine) when DT was employed as the photocatalyst. Yield and selectivity of o-nitroaniline and o-phenylenediamine employing either T or DT samples are displayed in Figure 2a,b, respectively. In the case of T sample, o-nitroaniline is only selective as a result of hydrogenation of o-dinitrobenzene (Figure 2a). The yield and selectivity boost with the increase of photoirradiation time reaching ~88.5% within 13 h (see Figure 2a). On the other hand, DT photocatalyst shows a higher yield and selectivity ~97% of o-phenylenediamine as a result of reduction and hydrogenation of the two-nitro groups of the o-dinitrobenzene after only 9 h irradiation as displayed in Figure 2b. By comparison, the commercial P25 $TiO_2$ was tested for photocatalytic reduction of o-dinitrobenzene under solar simulator light. The results showed the formation of non-selective reduction products from ≈13% o-nitroaniline and ≈86.5% o-phenylenediamine within 13 h under solar simulator light as displayed in Figure 2c. Irradiation of aqueous solution containing o-dinitrobenzene with $TiO_2$ samples and oxalic acid under solar light produced o-nitroaniline and o-phenylenediamine as a reduction product. On the other hand, no reduction products were obtained from the aqueous solution containing o-dinitrobenzene without using photocatalyst or light and/or oxalic acid, respectively. This means that these parameters are essential for reduction of o-dinitrobenzene to the corresponding o-phenylenediamine.

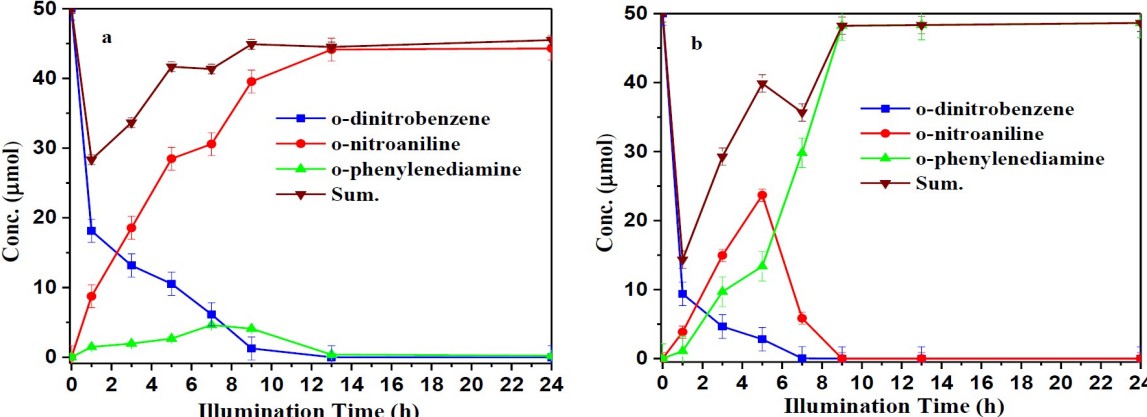

**Figure 1.** Time-dependent change in the concentration of substrate and products in aqueous solution of (**a**) un-doped $TiO_2$ (T sample) Note Conc.: Concentration and (**b**) (C, N, S) -doped $TiO_2$ (DT sample) in the presence of oxalic acid as hole scavenger during photoirradiation under simulated solar light, Note Conc.: Concentration and reaction conditions: 25 mg $TiO_2$ samples, 50 μmol o-dinitrobenzene, 250 μmol oxalic acid, 5 cm$^3$ deionized water, Ar.

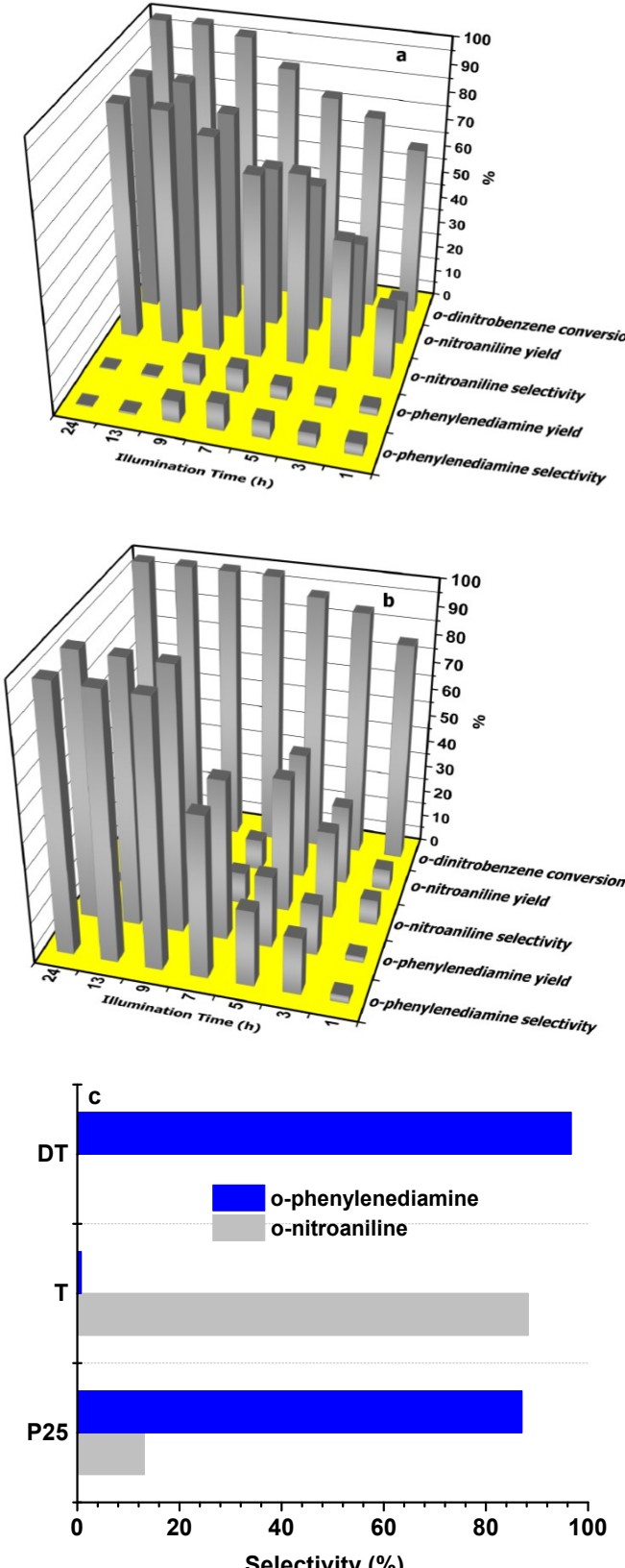

**Figure 2.** Yield and Selectivity for (**a**) T, (**b**) DT samples and (**c**) Selectivity for P25 compared with other samples, reaction conditions: 25 mg $TiO_2$ samples, 50 μmol o-dinitrobenzene, 250 μmol oxalic acid, 5 $cm^3$ deionized water, Ar.

It is well known that the light-induced six-electron reduction of a one-nitro group of the nitroaromatic compound to the corresponding amino compound in the presence of $TiO_2$ occurs via a sequence of electron transfer, protonation, and dehydrogenation reactions [18]. Thus, the complete reduction of two-nitro groups to two-amino groups requires twelve electrons and twelve protons. This usually occurs via the formation of hydroxylamine and/or nitrosobenzene as intermediates. However, neither nitrosobenzene nor N-phenyl hydroxylamine were detected in our cases. This might be explained by the fact that DT photocatalyst expedites the conversion of the nitro-to-amine through hydrogenation reactions (i.e., via hydrogen species derived from oxalic acid). This inhibits side reactions and facilitates selective o-phenylenediamine production. Therefore, with DT sample, photoirradiation leads to complete transformation of o-dinitrobenzene to the corresponding o-phenylenediamine with high yield and selectivity. It is also important to mention that the reduction of the second nitro group is usually more difficult that the first one and therefore it requires stronger reducing agent. The doped anatase/brookite biphasic $TiO_2$ (DT sample) showed the high ability to complete the reduction of the two-nitro group of the dinitro compound to diamino compound (o-phenylenediamine). Therefore, compared with commercial P25 $TiO_2$, un-doped and doped samples formed a selective reduction product from o-nitroaniline and o-phenylenediamine, respectively.

The observed difference in the selectivity of the photocatalytic conversion of o-dinitrobenzene employing the un-doped and doped materials can be attributed to the following different factors:

Firstly, this can be accredited to decrease in the band gap for the doped sample (2.9 eV) compared to the un-doped one (3.1 eV) 1 to enhancement its absorption capacity under solar simulator light (see UV-Vis. analysis, Figure S1) [17]. By non-metal doping, the O2P orbitals of $TiO_2$ mixes with the dopants 2P orbitals of C, N and S forming a new mid-gap above the valence band of $TiO_2$ (see Scheme 1, see XPS analysis, Figure S2) which leads to decrease its band gap. Briefly, as illustrated in our previous work [17], XPS analysis revealed C, N and S are doped with $TiO_2$ and carbon is also located on the surface (see Figure S2). Figure 2a illustrates the presence peaks of S2p with binding energy located at 168.5 eV for $S^{6+}$. Besides, Figure S2b displays the N1s peaks for the doped sample. It is clear that there are two constituent peaks at around 399.7 and 401.8 eV, without the peak at 396–397 eV definitely assigned to substitutional nitrogen. In the meantime, a peak observed at around 401 eV was credited to interstitial N-doping. Moreover, non-metal dopants lead to formation of $Ti^{3+}$ defects. This is due to the charge difference between N (-3) and O (-2) when N atoms bonded to Ti atoms [19]. The different electronic interactions of Ti with N anions may result in partial electron transfer from the N to Ti which may form $Ti^{3+}$ defects. The formation of these $Ti^{3+}$ species was verified using XPS and electron paramagnetic resonance (EPR) analyses (see Figures 3 and 4). From XPS analysis of the doped sample, the $Ti_{2p}$ spectrum revealed a slight negative shift of the two peaks at 457.7 eV ($Ti2p_{3/2}$) and 463.4 eV ($Ti2p_{1/2}$) with respect to $Ti^{4+}$ (458 eV, $Ti2p_{3/2}$ and 463.7 eV, $Ti2p_{1/2}$) in un-doped sample (see Figure 3). This shift revealed the formation of $Ti^{3+}$ species [20]. This new $Ti^{3+}$ species/ defects also enhance the electronic states for $TiO_2$ by the formation of isolated defect energy level from $Ti^{3+}$ below the bottom of conduction band for $TiO_2$ as displayed in Scheme 1 [21]. From EPR analysis, for the doped DT sample, the resonances at g values of less than 2.0 (1.96–1.92) are attributed to photogenerated electrons stabilized in Ti cations located at crystallization defects as shown in Figure 4. These trapped electrons could reduce $Ti^{4+}$, cause the formation of $Ti^{3+}$ paramagnetic species [22]. In general, the surface $Ti^{3+}$ has considerably lesser g factors value than those found in bulk. Additionally, the signal shapes for surface $Ti^{3+}$ is commonly broad, but in the inner (bulk) $Ti^{3+}$ has a narrow axially symmetric signal [23]. Thus, the g value of 1.92 was credited to the surface $Ti^{3+}$ species. Moreover, the g value of 1.943 and 1.961 was associated with the formation of bulk $Ti^{3+}$ [24–26]. Therefore, from XPS and EPR analyses, we can assume that the band gap for the doped sample decreased not only by non-metal dopants (C, N, and S) but also via formation of $Ti^{3+}$ defects. Subsequently, all these new-formed bands lead to enhancement of the absorption capacity of the doped sample under solar simulator light. Consequently, this leads to enhancing the photocatalytic activity of the doped sample for complete reduction of o-dinitrobenzene to the corresponding o-phenylenediamine.

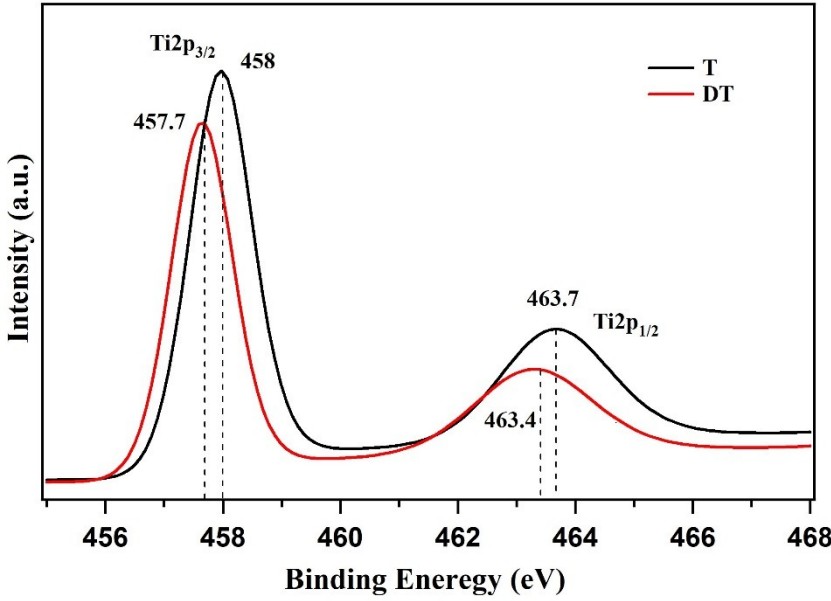

**Figure 3.** XPS detailed scans in the energy regions of Ti2p for T and DT samples.

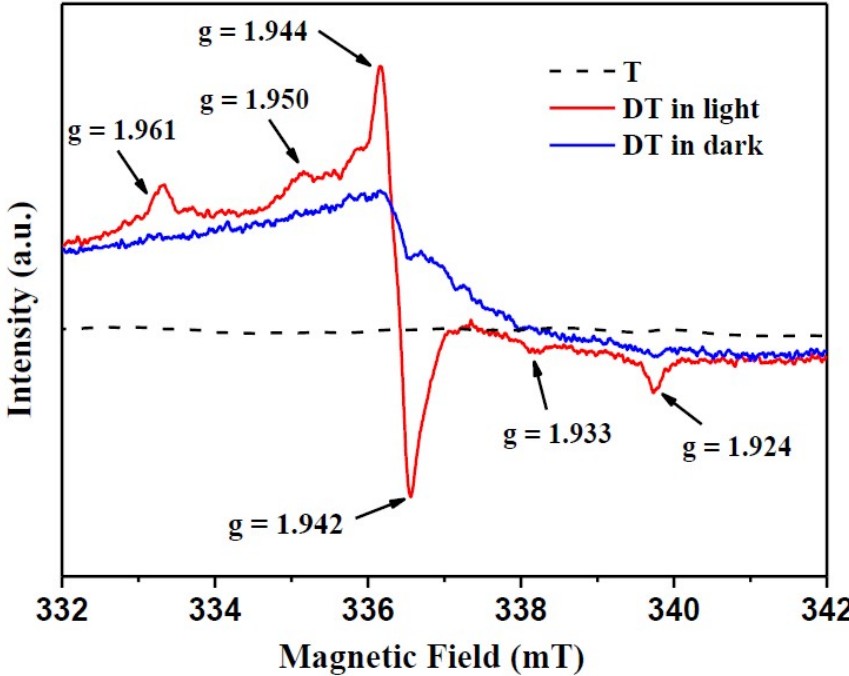

**Figure 4.** EPR spectra of T and DT samples, the DT sample recorded at dark and under UV irradiation (after 5 min) at room temperature. Instrument setting: operating at 9.41 GHz field modulation. modulation amplitude: 0.2 mT, power: 10 mW, gain: 5.

Secondly, these new electronic states act as electron-hole traps which leads to an increase in their lifetime by reducing the electron-hole recombination, resulting in an enhancement of the photocatalytic activity for the doped sample. The lifetimes and charge carrier trapping can by determined using laser flash photolysis [27]. The absorption time profile noticed at the selected wavelength (600 nm) for the un-doped and doped samples is shown in Figure 5. It can be clearly noticed that the initial decay for the un-doped T sample is faster than that of the doped DT sample. This can be attributed to presence of the long-lived $Ti^{3+}$ species. Moreover, the amount of the generated charge carriers upon irradiating the doped sample is higher than that formed in the un-doped one. Overall, the doped sample demonstrates the utmost significant charge generation and the maximum number of hole–electron

pairs available to participate in surface redox reactions with adsorbed species. This finding is in very good agreement with the photoreactivity results, as the above doped sample seemed to be the most photoactive for the studied photoreduction reaction. Combining the results of XPS, EPR and TAS analyses, it can be deduced that the non-metal dopants caused the formation of surface/ bulk $Ti^{3+}$ in DT sample. Therefore, this new defect results in an enhancement in the absorption capacity of the material for complete the reduction of o-dinitrobenzene to the corresponding o-phenylenediamine under solar simulated light. Another evidence on the effect of doping on the charge carrier's lifetime can be gained from the PL analysis. Figure 6 shows the PL spectra of T and DT samples at excitation wavelength (259 nm) using Xe lamp at room temperature for further evidence out finding results. The PL spectra for both samples are similar with different intensities. The PL spectra of T and DT samples revealed several emission peaks, the maximum and centered one at about 470 nm, which were referred to as the shallow energy level excitonic PL phenomenon [28]. Moreover, we can notice that the PL intensity of the doped sample decreased compared to the un-doped one. The lower PL signals for the doped sample may indicate the lower electron-hole recombination rate and the higher separation efficiency and this result agrees with TAS measurements. From the above, it can be concluded that the enhancement of the photocatalytic activity of the doped sample using solar light irradiation is not only due to the formation of a mid-gap level via non-metal dopants (C, N, S) above the valence band, but also due to the formation of isolated defect energy level ($Ti^{3+}$) below the bottom of the conduction band of doped sample. This finding leads to a decrease of the band gap, decrease of the charge recombination, and increase of the life time of the charge carriers for the doped sample and consequently leads to enhancement of the complete reduction of o-dinitrobenzene to the corresponding o-phenylenediamine as shown in Scheme 1. Thirdly, hydrogen species (maybe associated with $Ti^{3+}$) derived from oxalic acid facilitates the complete reduction of o-dinitrobenzene (see Scheme 1). Moreover, the formed $Ti^{3+}$ atoms act as active sites for the reduction of o-dinitrobenzene and o-nitroaniline to the corresponding o-phenylenediamine. These surface $Ti^{3+}$ atoms behave as an adsorption site for o-dinitrobenzene and o-nitroaniline via an electron donation and as a trapping site for photogenerated electron formed in conduction band (see Scheme 1) [15,16,29]. Therefore, these avenues facilitate the achievement of the reduction process of the two-nitro group of o-dinitrobenzene to the corresponding o-phenylenediamine. Finally, the high crystallinity and the mesoporosity leads to improvement of the photocatalytic activity of the doped sample (see Figure S3) [17]. This attribute to the doped sample was calcined at 450 °C. Up to calcination, the organic remains in $TiO_2$ matrix disintegrated and formed a highly mesoporous material with pore-size diameter 3.6 nm compared with the un-doped T sample with pore-size diameter about 2.2 nm [17]. This mesoporous nature for the doped sample facilitated the adsorption capacity of the nitro aromatic compounds. Consequently, this enhanced the photocatalytic activity of the doped sample for complete reduction of o-dinitrobenzene under solar stimulator light. For all the above-mentioned reasons, it can be shown that the doped sample has versatile properties and great ability for highly selective photocatalytic reduction of o-dinitrobenzene to the corresponding o-phenylenediamine under solar simulator light.

On the other hand, one of the essential parameters in photocatalytic applications in an aqueous medium is the stability and reusability of the prepared samples at the end of the reaction. The stability of the doped sample can be investigated by monitoring the UV-Vis analysis at the end of the reaction. The results show that there is no change in the reflectance behavior of the doped sample as shown in Figure S1. This clearly revealed DT sample has good stability. Furthermore, the reusability for the doped sample was investigated after four cycles (Figure 7). The o-phenylenediamine yield is slightly decreased owing to a little amount of photocatalyst loss during product separation.

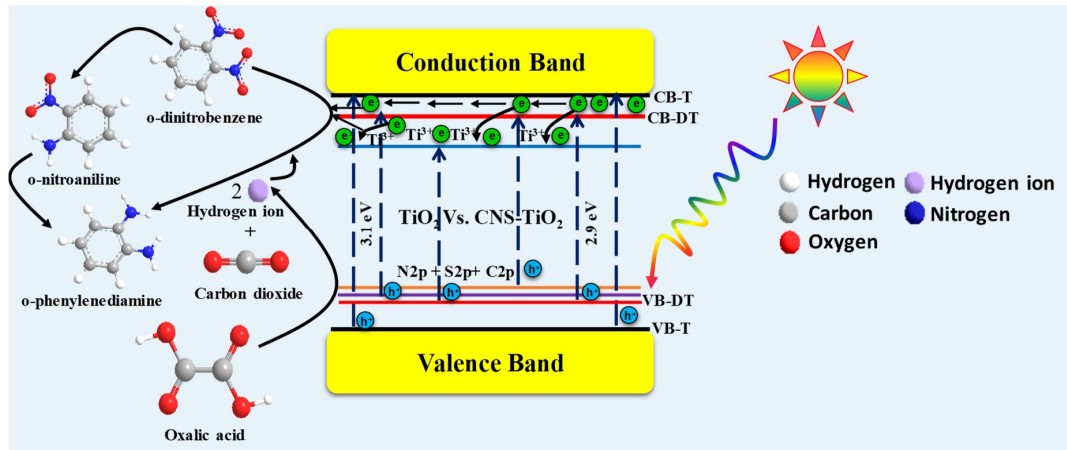

**Scheme 1.** Suggested mechanism for the effect of non-metal dopants (C, N, S) and $Ti^{3+}$ surface defects in the photocatalytic conversion of o-dinitrobenzene to the corresponding o-phenylenediamine under solar light.

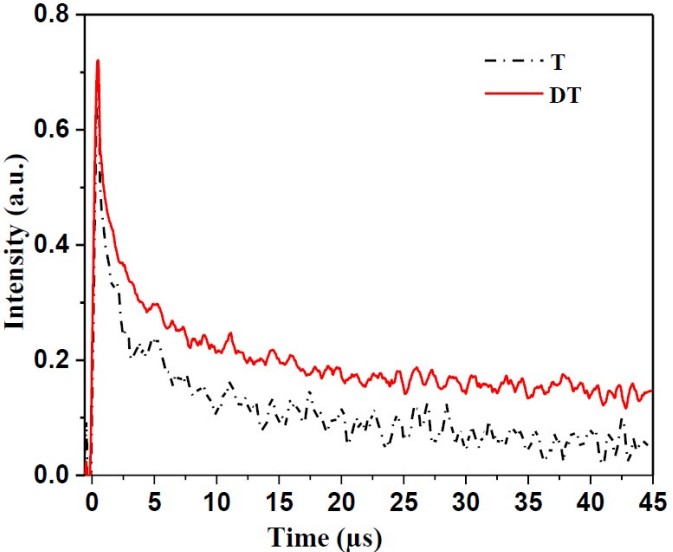

**Figure 5.** Absorption time profile of T and DT samples at 600 nm.

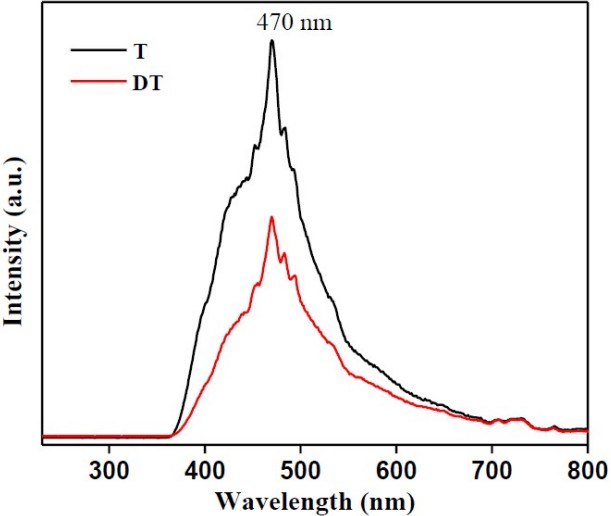

**Figure 6.** PL spectra of T and DT samples.

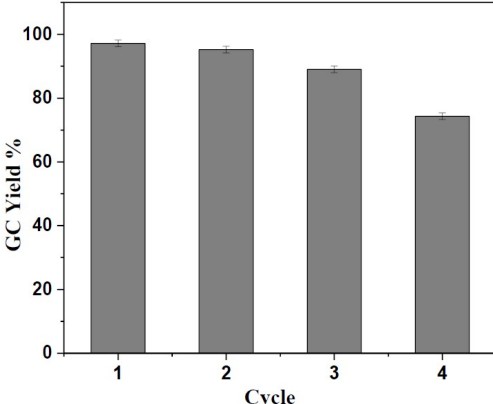

**Figure 7.** Reusability and photocatalytic efficiency of the DT aqueous solution for reduction of o-dinitrobenzene to corresponding o-phenylenediamine under solar simulated light irradiation after 24 h, reaction conditions: 25 mg DT sample, 50 µmol o-dinitrobenzene, 250 µmol oxalic acid, 5 cm$^3$ deionized water, Ar Experimental section.

## 2.2. Materials and Chemicals

The triblock copolymer surfactant poly (ethylene glycol)-poly (propylene glycol)- poly (ethylene glycol) (P-123, M wt. ~5800), titanium tert-butoxide $Ti(OC(CH_3)_3)_4$ (TBOT), thiourea (≥99%), Triton-X 100, polyethylene glycol (10,000 MW), sodium sulfate (>99%), oxalic acid dihydrate (≥99%), dichloromethane (High-performance liquid chromatography (HPLC) grade, >99.9%), ethanol (99.8%), o-dinitrobenzene (≥99%), o-nitroaniline (98%) and o-phenylene diamine (99%) were purchased from Sigma-Aldrich, Darmstadt, Germany and were used as received.

## 2.3. Photocatalysts Preparation

The preparation procedure of un-doped anatase/brookite biphase $TiO_2$ and (C, N, S)-doped anatase/brookite biphase $TiO_2$ was published [17]. Un-doped anatase/brookite biphase $TiO_2$ was produced via sol-gel method using TBOT as a $TiO_2$ source and P123 as a directing agent. Then, the prepared $TiO_2$ powder was mixed with thiourea in a weight ratio of 1:1 and calcined in a covered vessel at 450 °C for 1 h to get (C, N, S) -doped anatase/ brookite biphase $TiO_2$. The obtained samples were donated as T and DT for un-doped anatase/ brookite biphase $TiO_2$ and (C, N, S)-doped anatase/ brookite biphase $TiO_2$.

## 2.4. Sample Characterization

EPR spectra were recorded at room temperature on a MiniScope X-band EPR spectrometer (MS400 Magnettech GmbH, Berlin, Germany) operating at 9.41 GHz field modulation. modulation amplitude: 0.2 mT, power: 10 mW, gain: 5. The experimental EPR spectra acquisition and simulation were carried out. The surface chemical composition of the samples was determined using X-ray Photoelectron Spectroscopy, Thermo Fisher Scientific K-Alpha XPS system (Waltham, MA, USA) with X-ray source –Al Ka micro-focused mono-chromator. The binding energies of surface adventitious carbon calibrated to the C1s peak at 284.4 ± 0.1 eV. Spectrofluorophotometer (RF-5301 PC, Shmidzu, Tokyo, Japan) was used to determine the photoluminescence (PL) spectra of the samples at room temperature with excitation wavelength 259 nm. Nanosecond diffuse reflectance transient absorption spectroscopy measurements were performed using an experimental set-up as reported previously [30]. For measurements, all powders were purged for $\frac{1}{2}$ h with $N_2$ prior to the measurements.

## 2.5. Photocatalytic Reaction Procedure

The photocatalytic reactions were carried out in a sealed glass snap-cap bottle (23 mm in diameter and 75 mm in length) with contentious stirring. 25 mg $TiO_2$ (un-doped or doped or P25) were

suspended in 5 cm$^3$ of deionized water containing 50 μmol of the o-dinitrobenzene and ~250 μmol oxalic acid. The mixture was stirred in the glass snap-cap bottle in the dark with Ar being purged for 15 min to remove molecular oxygen. Then the mixture was irradiated for 24 h using solar simulator (SOL1200 lamp, UV (A) was measured by *Dr.* K *Hönle* UV (A)-detector (Munich, Germany) to be *20 mW/cm$^2$*). Afterward, the excess amount of oxalic acid was neutralized by adding desired amount of $NH_4OH$ followed by extraction of the reactant and products from the aqueous phase by dichloromethane to be quantitatively and qualitatively analyzed by Gas Chromatograph-Mass Spectrometry (GC/MS) and GC with Flame Ionization Detector (GC-FID), respectively, after filtration through 0.2 μm filter. Shimadzu GC/MS-QP 5000 (Tokyo, Japan) equipped with a 30 m Rxi-5ms (*d* = 0.32 mm) capillary column with operating temperatures programmed: injection temperature 310 °C, oven temperature 120 °C (hold 2 min) from 120 to 280 °C at a rate of 10 °C min$^{-1}$, 280 °C (hold 15 min) in splitless mode, injection volume was 3.0 μL with helium as a carrier gas was used to qualitative analysis. Shimadzu GC 2010 (Tokyo, Japan) equipped with a Rtx-5 (*d* = 0.25 mm) capillary column and an FID detector was used to define the concentration of the reactant and of the products. Operating temperatures programmed: injection temperature 250 °C, oven temperature 70 °C (hold 2 min) from 70 to 280 °C at a rate of 10 °C min$^{-1}$, in splitless mode. Injection volume was 2.0 μL with nitrogen as the carrier gas. The concentrations of the reactant, besides the products, were evaluated according to the calibration curves prepared with authentic standards.

## 3. Conclusions

Mesostructured anatase-brookite biphase un-doped $TiO_2$ and (C, N, S) doped anatase-brookite biphase $TiO_2$ photocatalysts have various selectivities towards the reduction of o-dinitrobenzene in aqueous solution in the presence of oxalic acid as a hole scavenger under solar simulator light. Compared with commercial P25 $TiO_2$, the un-doped material showed a good selectivity (85.5%) towards the reduction of just the one-nitro group, i.e., towards the production of o-nitroaniline. On the other hand, (C, N, S) the doped sample displayed a high selectivity (97%) towards the complete reduction of the two-nitro group in o-dinitrobenzene to the corresponding o-phenylenediamine. The superiority of the doped $TiO_2$ photocatalysts is attributed to the formation of new dopants (C, N, S) as hole traps, the formation of $Ti^{3+}$ defects and increase in the lifetime of the charge carriers, which leads to enhancement of the absorption capacity under solar simulator light. Furthermore, the surface $Ti^{3+}$ atoms of doped $TiO_2$ act as the adsorption site for nitroaromatics and the trapping site for photogenerated electrons formed on the conduction band. This finding leads to the acceleration of rapid nitro-to-amine reduction/hydrogenation and the complete formation of o-phenylenediamine, while suppressing side reactions.

**Supplementary Materials:** The following are available online at http://www.mdpi.com/2073-4344/8/12/641/s1, Figure S1: (a) UV-Vis absorption spectra (b) Tauc plots of modified Kubelka-Munk function of samples T, and DT before and after reusing., Figure S2: XPS detailed scans in the energy regions of (a) S2p, (b) N1s and (c) C1s for DT sample, Figure S3: (a) Low angle XRD patterns and (b) Wide angle XRD patterns for the T and DT samples.

**Author Contributions:** Conceptualization, H.M.E.-H., S.M.E.-S., A.H. and A.A.I.; methodology, H.M.E.-H.; formal analysis and data curation, H.M.E.-H., A.H. and S.M.E.-S.; writing—original draft preparation, H.M.E.-H., S.M.E.-S. and A.A.I.; writing—review and editing, H.M.E.-H., R.D., S.M.E.-S., A.H. and A.A.I.; resources, D.W.B. and S.M.E.-S.; supervision, H.M.K., I.A.I. and D.W.B.

**Funding:** This work was supported by short cycle 5, Science & Technology Development Fund in Egypt (STDF fellowship) under Grant no. ID 12282.

**Acknowledgments:** This work was supported by short cycle 5, Science & Technology Development Fund in Egypt (STDF fellowship) under Grant no. ID 12282. H. El-Hosainy acknowledges Institut für Technische Chemie, Leibniz Universität Hannover, Germany for hosting him during the current research work. The publication of this article was funded by the Open Access Fund of the Leibniz Universität Hannover.

**Conflicts of Interest:** The authors declare no conflicts of interest.

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
