# Peer review of "Highly Selective Photocatalytic Reduction of o-Dinitrobenzene to o-Phenylenediamine over Non-Metal-Doped TiO2 under Simulated Solar Light Irradiation"

_catalysts, doi:10.3390/catal8120641_

Round 1
Reviewer 1 Report
The manuscript titled “Highly selective photocatalytic reduction of o-dinitrobenzene to o-phenylenediamine over non metals dopes TiO2 under simulated solar light irradiation” presents some interesting points and discussion towards a more effective and green method to drive chemical reactions by using solar light. However, the authors have not included all the data needed to support these claims.
Point 1. English of the manuscript must be massively improved. In addition, the manuscript is not easy to follow and therefore, it must be improved.
Point 2. All figures in the manuscript do not show a caption. Captions must be added to every single figure.
Point 3. The authors present a biphasic anatase-brookite (C, N, S) doped photocatalysts, however, there is no XPS data to support this claim. XPS data must be shown in the main txt as it is extremely important to support all presented claims. In addition, the authors mention that their samples present nitrogen in a substitutional position. However, N-substitutional is shown at 396 eV and not at 400 eV, which is attributed to interstitial/ and or adsorbed nitrogen (N0). As previously mentioned, XPS data and a deeper discussion of the XPS findings must be included in the main text.
Point 4. The authors mention the presence of Ti3+ defects as detrimental factors to enhance the lifetime of the photogenerated charge carriers. However, there is no XPS data to support (corroborate) this claim.
Point 5. The authors mention changes in the bandgap and optical properties upon doping of the biphasic. However, despite that the authors mention that the UV-Vis data is shown in the supporting information, there is not supporting information attached to the manuscript. I would strongly recommend incorporating this data to the main text, as it would be important for the readers of this work.
Point 6. The authors mention that a scheme to understand the mechanism of the photocatalytic reduction of o-dinitro benzene is included within the text. However, this scheme (Scheme 1) is missed from the manuscript.
Point 7. The main topic from this work is the photocatalytic activity of the as-prepared samples. However, the photocatalysis section of the manuscript consist of only 7 lines. I would strongly recommend expanding this section and have a deeper discussion of the photocatalytic activity of the materials presented in this work. In addition, important data, showing the photocatalytic reduction of o-dinitrobenzene, is being missed from the manuscript, as only the cycling data is shown in this section. The authors also claim a clear enhancement of the as-synthesised materials compare to P25, however, they have not provided any data (graphs) to support this claim. Therefore, a deeper discussion and supporting data must be included in the main text.
In conclusion, I would not recommend this work for publication in Catalysis journal until all the above points are fully addressed by the authors.
Author Response
The manuscript titled “Highly selective photocatalytic reduction of odinitrobenzene to o-phenylenediamine over non-metals dopes TiO2 under simulated solar light irradiation” presents some interesting points and discussion towards a more effective and green method to drive chemical reactions by using solar light. However, the authors have not included all the data needed to support these claims.
Point 1. English of the manuscript must be massively improved. In addition, the manuscript is not easy to follow and therefore, it must be improved.
We have carefully read and correct the English, misleading statements, missing spaces and typographical errors. The revised manuscript has been thoroughly edited and modified the language and presentation.
Point 2. All figures in the manuscript do not show a caption. Captions must be added to every single figure.
We have added all figure captions in the revised version.
Point 3. The authors present a biphasic anatase-brookite (C, N, S) doped photocatalysts, however, there is no XPS data to support this claim. XPS data must be shown in the main txt as it is extremely important to support all presented claims. In addition, the authors mention that their samples present nitrogen in a substitutional position. However, N-substitutional is shown at 396 eV and not at 400 eV, which is attributed to interstitial/ and or adsorbed nitrogen (N0). As previously mentioned, XPS data and a deeper discussion of the XPS findings must be included in the main text.
The reviewer is certainly correct, in this manuscript, we have focused our efforts on the rather fundamental question what effect anatase-brookite biphasic TiO2 and non-metals doped anatase-brookite biphasic TiO2 photocatalysts on the photocatalytic reduction and hydrogenation reaction of o-dinitrobenzene in the presence of oxalic acid over under solar simulated. For this purpose, we have now compared non-doped TiO2 and doped one. On the other hand, we added XPS graphs for C, N and S peaks in a supporting information and its discussion in the main text (Line 239-248). This is due to these graphs were already published in our previous work[1]. On the other hand, as you mentioned our XPS spectra shown two constituent peaks at around 399.7 and 401.8 eV, without the peak at 396-397 eV definitely assigned to substitutional nitrogen. So, the peak observed at around 401 eV were credited to interstitial N-doping not for N- substitutional doping as you mentioned. Thus, we removed this mistake from main text and added new sentences (Line 242-248). So, we are so sorry for this mistake.
1. El-Sheikh, S. M. et al., J. Hazard. Mater. 2014, 280, 723–733.
Point 4. The authors mention the presence of Ti3+ defects as detrimental factors to enhance the lifetime of the photogenerated charge carriers. However, there is no XPS data to support (corroborate) this claim
We acknowledge these valuable comments which certainly lead to the increase of the impact of this current paper. We have added XPS data for Ti2p (Figure 3) with complete discussion to prove the presence of Ti3+ in TiO2 lattice (Line 250-254).
Point 5. The authors mention changes in the bandgap and optical properties upon doping of the biphasic. However, despite that the authors mention that the UV-Vis data is shown in the supporting information, there is not supporting information attached to the manuscript. I would strongly recommend incorporating this data to the main text, as it would be important for the readers of this work.
Actually, supporting information missed in our initial submission. Now, we attached it in the revised version. Herein, we added UV-VIS in supporting information due to we already published this graphs before in our previous work as we mentioned before [1] in term of photodegradation of toxic organic compounds. Meanwhile, we need to mention that this works is continuation of our previous work published in J. Hazard. Mater. 2014, 280, 723-733, as in both cases we have got the same photocatalyst: un-doped and doped anatase-brookite biphasic TiO2. But here, we report for the first time, these well-defined materials for the high selective hydrogenation of o-dinitrobenzene to the corresponding o-nitroaniline and o-phenylenediamine under solar simulator light, respectively.
Point 6. The authors mention that a scheme to understand the mechanism of the photocatalytic reduction of o-dinitro benzene is included within the text. However, this scheme (Scheme 1) is missed from the manuscript.
Thank you so much for your notification and suggestion. Now, we added scheme 1 in our manuscript (Line 368-378).
Point 7. The main topic from this work is the photocatalytic activity of the as prepared samples. However, the photocatalysis section of the manuscript consist of only 7 lines. I would strongly recommend expanding this section and have a deeper discussion of the photocatalytic activity of the materials presented in this work. In addition, important data, showing the photocatalytic reduction of o-dinitrobenzene, is being missed from the manuscript, as only the cycling data is shown in this section. The authors also claim a clear enhancement of the as-synthesised materials compare to P25, however, they have not provided any data (graphs) to support this claim. Therefore, a deeper discussion and supporting data must be included in the main text
We have added some sentences to discuss the difference in photocatalytic activity of the prepared photocatalyst in deep from Line 235 to Line 364. On the other hand, the important data for the photocatalytic reduction of o-dinitrobenzene was displayed on Figure 1 and Figure 2. Moreover, we have added Figure 2c to prove the clear enhancement of our TiO2 synthesized materials for complete the photocatalytic reduction of o-dinitrobenzene under solar light compared to P25.
The manuscript has been revised according these valuable and deep contributions of the reviewer.
Reviewer 2 Report
The article entitled "Highly selective photocatalytic reduction of o-dinitrobenzene to o-phenylenediamine over non-metals doped TiO2 under simulated solar light irradiation" is the natural continuation of the article visible in J. Hazard. Mater. 2014, 280, 723-733, as in both cases we have got the same photocatalyst: un-doped and doped anatase-brookite biphasic TiO2.I do not think this is something incorrect, on the contrary, it's good that the authors are looking for new applications for structurally defined materials.The fact is that in order to understand this article, it is necessary to read the earlier one, because there is the entire photocatalyst characterization - which is done very minutely.
I have no objection to this manuscript except that it is necessary to add descriptions to the drawings, because in their present form they are incomprehensible.
I can not see Scheme 1, Figure S1 and S2 - maybe I can not find them, but I think they were not included to the manuscript - it should be checked.
English language and style are minor spell check required
Author Response
The article entitled "Highly selective photocatalytic reduction of odinitrobenzene to o-phenylenediamine over non-metals doped TiO2 under simulated solar light irradiation" is the natural continuation of the article visible in J. Hazard. Mater. 2014, 280, 723-733, as in both cases we have got the same photocatalyst: un-doped and doped anatase-brookite biphasic TiO2.I do not think this is something incorrect, on the contrary, it's good that the authors are looking for new applications for structurally defined materials. The fact is that in order to understand this article, it is necessary to read the earlier one, because there is the entire photocatalyst characterization - which is done very minutely.
I have no objection to this manuscript except that it is necessary to add descriptions to the drawings, because in their present form they are incomprehensible. I cannot see Scheme 1, Figure S1 and S2 - maybe I cannot find them, but I think they were not included to the manuscript - it should be checked.
Thank you so much for your time for reviewing our paper.
Actually, scheme 1 (Line 368-378) and supporting information were missed in our initial submission. Now, we added scheme 1 (Line 368-378) in the revised version and supporting information (Figure S1, S2 and S3). Also, figure captions were inserted in the text
English language and style are minor spell check required
We have carefully read and correct the English, misleading statements, missing spaces and typographical errors. The revised manuscript has been thoroughly edited and modified the language and presentation.
We have done our best to revise the manuscript following the valuable suggestions of both reviewers for which we are really grateful. We hope that the revised manuscript will now be acceptable for publication in the journal
Round 2
Reviewer 1 Report
The manuscript titled “Highly selective photocatalytic reduction of o-dinitrobenzene to o-phenylenediamine over non metals dopes TiO2 under simulated solar light irradiation” presents some interesting points and discussion towards a more effective and green method to drive chemical reactions by using solar light. The authors have addressed all comments suggested in the first review of this work. Therefore, I would recommend this work for publication in Catalysis journal.